# Artificial Intelligence-Driven Ensemble Model for Predicting Mortality Due to COVID-19 in East Africa

**DOI:** 10.3390/diagnostics12112861

**Published:** 2022-11-18

**Authors:** Kedir Hussein Abegaz, İlker Etikan

**Affiliations:** 1Biostatistics and Health Informatics, Public Health Department, Madda Walabu University, Robe 247, Ethiopia; 2Department of Biostatistics, Faculty of Medicine, Near East University, Near East Avenue, North Cyprus, Mersin 10, Nicosia 99138, Turkey; 3HOD Biostatistics, Faculty of Medicine, Near East University, Near East Avenue, North Cyprus, Mersin 10, Nicosia 99138, Turkey

**Keywords:** COVID-19, artificial intelligence, ensemble model, FFNN, ANFIS, SVM, East Africa

## Abstract

East Africa was not exempt from the devastating effects of COVID-19, which led to the nearly complete cessation of social and economic activities worldwide. The objective of this study was to predict mortality due to COVID-19 using an artificial intelligence-driven ensemble model in East Africa. The dataset, which spans two years, was divided into training and verification datasets. To predict the mortality, three steps were conducted, which included a sensitivity analysis, the modelling of four single AI-driven models, and development of four ensemble models. Four dominant input variables were selected to conduct the single models. Hence, the coefficients of determination of ANFIS, FFNN, SVM, and MLR were 0.9273, 0.8586, 0.8490, and 0.7956, respectively. The non-linear ensemble approaches performed better than the linear approaches, and the ANFIS ensemble was the best-performing ensemble approach that boosted the predicting performance of the single AI-driven models. This fact revealed the promising capability of ensemble models for predicting the daily mortality due to COVID-19 in other parts of the globe.

## 1. Introduction

“Artificial Intelligence could be the saviour of the COVID-19 pandemic in the coming year; we just need to prove it.”*The Lancet Digital Health*, 2021

Most pandemics in the 20th and 21st centuries were caused either by the coronavirus or the influenza virus. The coronavirus disease 2019 (COVID-19) is one of the 21st-century pandemics and highly contagious infections caused by severe acute respiratory syndrome coronavirus 2 (SARS-CoV2) [1,2]. The WHO, on 11 February 2020, named this outbreak COVID-19, used as a shorthand for coronavirus disease 2019. Again, the WHO, on 30 January 2020, affirmed this outbreak as a “Public health emergency of international concern” and finally as a “Pandemic” on 11 March 2020 [3].

In terms of mortality, COVID-19 has caused more than 6.5 million deaths (6,559,902 as of 8 October 2022) globally, with a case fatality rate of 2.04% [4]. This number proves that the pandemic is much different, in terms of global crises, compared with previous flu pandemics, such as the Spanish flu (in 1918), the Asian flu (1957–1958), the Hong Kong flu (1968–1970), and the swine flu (2009–2010). The nature of this pandemic made COVID-19 the first global public health issue that had a brutal impact on the global economy, which triggered a near to total shutdown of social and economic activities. Finally, the pandemic has shrunk the global economy by nearly 3 per cent according to the prediction of the International Monetary Fund (IMF) [5].

Remarkably, the crisis due to COVID-19 proves that our earth is unprepared for such a quickly spreading and rampant virus, resulting in a catastrophic pandemic. In addition to this, the big question, then, is “when will things go back to normal, or whether we should prepare for new waves of coronavirus or not?” Though no one has a final answer to this question, through data analyses, we can understand how it happened and what the situation will look like in the future. The results of these analyses, including those using artificial intelligence (AI)-driven models, will be actionable knowledge that can help us to manage a similar crisis in the future [6,7].

In this catastrophic era, AI is contributing the development of many effective strategies that can control the infection in real time and easily track the rampant virus [3]. It is also successfully used for the identification of the disease, monitoring of cases and deaths, and prediction of future outbreaks and risks of mortality by analyzing the previous data of patients in regard to the cases and deaths. In addition, AI can significantly boost the consistency of treatment and decision making by developing important data-driven algorithms [8,9].

Furthermore, web applications were developed by Chowdhury D. et al. in 2022 that can detect whether a patient has COVID-19 or not after the image of the chest X-ray is uploaded to the web application [10]. Through AI-supported imaging technology, unenhanced chest computed tomography (CT) becomes applicable to the prediction of COVID-19. According to Sciaffino S. et al., in 2021, multilayer perceptron was the best-performing machine learning algorithm in predicting the pulmonary parenchymal and vascular damage using unenhanced chest CT [11].

As in other parts of Africa, the crisis of the COVID-19 pandemic in East Africa continues to influence people within and across the region. The crisis has adversely affected the economies of countries, and the impact was severe in some parts of the region [12]. The resulting crisis and the pandemic itself threaten to reverse the development of some parts of the region that occurred within the last decade and will hinder progress toward sustainable development growth (SDG) [12,13].

In East Africa, communicable diseases were the leading causes of mortality in the earlier stage of the COVID-19 pandemic, and among these diseases, perinatal, maternal, and malnutrition cases were responsible for almost half of the mortality in the region. In addition to this, East Africa is facing momentous health-related challenges due to preventable infectious diseases. However, it is still facing a challenge due to the pandemic, and mortality remains at an alarming rate [13]. This rate is likely to increase in the coming years because of COVID-19 and its consequences in the region. Between the start of the COVID-19 pandemic and 8 October 2022, the pandemic caused 38,137 deaths in the region, and this number provides evidence showing that the region accounts for 14.8% of deaths on the African continent, which witnessed 257,672 deaths [4].

Even though many studies [14,15,16,17,18,19] have produced information regarding COVID-19 by using the concept of big data, machine learning and artificial intelligence, studies related to the prediction of mortality using different AI-driven models are rare globally and in the region. Based on our search, no study has reported on the use of AI-driven ensemble modelling to predict mortality due to COVID-19 in the region. Hence, in response to the stated gap, this study proposed and aimed to select the best AI-driven model for predicting mortality due to COVID-19 in the region and to develop an AI-driven ensemble model that can be used to predict mortality due to COVID-19 in East Africa.

## 2. Materials and Methods

### 2.1. Study Area

East Africa is the most populous sub-region of Africa, representing nearly 5.6% of the world’s population. This region stretches from Mozambique in the south to Eritrea in the north. There are eighteen countries and two independencies in the region, but nearly a quarter of the region’s people are living in one country, Ethiopia [20,21].

### 2.2. Data Source and Attribute Selection

This study used COVID-19-related data collected daily over 24 months, from April 2020 to April 2022, in the region. These data were public data from the “Our World in Data (OWID)” team and the COVID-19 data warehouse at John Hopkins University (JHU), collected by the Center for Systems Science and Engineering (CSSE), which is open to researchers. The data were retrieved from: https://github.com/owid/COVID-19-data/tree/master/public/data (accessed on 25 June 2022). The OWID, in its statement under the license section, explains that “All visualizations, data, and code produced by ‘Our World in Data’ is completely open access under the Creative Commons BY license. You have the permission to use, distribute, and reproduce these in any medium, provided the source and authors are credited” [22,23]. Ethical approval was not necessary, as this dataset does not include personal information and is public, as approved by the CSSE and JHU. 

### 2.3. Data Preprocessing and Analyses

In the ‘Our World in Data (OWID)’ COVID-19 database, many variables are available, but ten variables were selected because of the completeness of their data and their relationship with mortality. Datasets were retrieved for each country in the East Africa region independently, and we calculated the average values to represent the region with a single variable for mortality and other input variables. It is known that data collected daily is non-linear by nature. Hence, the first step that we took was to normalize each variable in the dataset.

The second activity that we undertook was to select the dominant input variables through a non-linear sensitivity analysis called the coefficient of determination (DC). This analysis was conducted using an artificial neural network (ANN), applying one target and one input variable to predict the estimated values and calculate the coefficient of determination so as to verify the correlation of each input variable with the target variable.

The dataset from these countries was classified into two training (70%) and testing (30%) sub-datasets for the development of AI-driven single and ensemble models. For all the developed AI-driven models, the target variable was the daily number of new deaths due to COVID-19 in the region, and the input variables were the new daily number of cases in the region, the positive rate, the number of people vaccinated, hospital beds/1000 patients, and so on. In Table 1, the list of all the variables used in this study and their explanations are presented.

In the data processing step, the data normalization was calculated using Microsoft Excel. However, the menu-based MATLAB (Version 20) was applied to conduct the sensitivity analysis, the single black-box AI-driven models, and AI-driven ensemble models.

### 2.4. Proposed Methods

In this study, we modelled three AI-driven models, including an adaptive neuro-fuzzy inference system (ANFIS), feedforward neural network (FFNN), support vector machine (SVM), and one conventional data-driven model, multiple linear regression (MLR), to predict mortality due to COVID-19 in East Africa. In addition, we classified a training dataset and a test dataset after normalizing the inputs. Figure 1 shows that three stages were conducted to carry out the given study. Firstly, the selection of dominant inputs for the prediction of COVID-19 mortality in the region was conducted to rank and select the most influential input variables for the modelling. In the second stage, four AI-driven black-box models (ANFIS, SVM, FFNN, and MLR) were applied independently to predict the COVID-19 mortality. Thirdly, as a final stage, four ensemble approaches, namely the ANFIS ensemble (ANFISE), neural network ensemble (NNE), weighted average ensemble (WAE), and simple average ensemble (SAE), were constructed. In the ensemble stage, the estimated output of every single model was used as an input for the AI-driven ensemble process. Then, the predicted mortality based on the ensemble model was compared with the predicted results from each of the black-box models in the second stage.

#### 2.4.1. The Feedforward Neural Network (FFNN)

The artificial neural network (ANN) is one of the most significant AI-driven models, because it can build links between the target and input variables by training the neural network without having comprehensive information on the entire data set [24]. This model is a self-learning simulation function that demonstrates the capacity to model and forecast complicated processes. This capability makes ANN a more practical and efficient model in different domains of science, such as biomedical technology, engineering, agriculture, and business [25].

Because of its simplicity and favored ability to react to various challenges without considering the past information regarding the process, this study used the feedforward neural network (FFNN), employing propagation algorithms. FFNN is formed of linked pieces called ‘nodes’ that have unit properties of information, such as learning, nonlinearity, noise tolerance, generalization capability, and so on, and it has three layers (see Figure 2), including the input, the hidden, and the output layers. As a result, the input variables provided to the input layers’ ‘neurons’ are transmitted forwards, and the activation function, a non-linear function, is employed to construct the output vector.

A multi-layer perceptron (MLP) model with a single hidden layer was computed in this study. The formal definition of this model is as follows: the function *‘f*’ on the fixed-size input *‘x*’, such as *f(x) ≈ y* for training pairs of *(x, y).* Alternatively, recurrent neural networks learn sequential data, computing the output *‘Ø*’ on the variable-length input *Xk = {x1… Xk} ≈ yk* for training pairs of *(Xn, Yn)* for all *1 ≤ k ≤ n.*

In the definition of FFNN with the *‘m*’ layer (or *‘m-2′* hidden layers) prototype, the output perceptron has an activation function *Øo*, and the hidden layer perceptron has activation functions *Ø*. Every perceptron in layer li is connected to every perceptron in layer *l_i−1_*_._ The layers are fully connected, and there is no connection between the perceptrons in the same layer. Hence, According to Brilliant [26], it is computed using the following formulas:

**First**, initialize the input layer *l*_0_ and set the values of the outputs ${Ø}_{i}^{0}$ for nodes in the input layer *l*_0_ in relation to their associated inputs in the vector $\vec{x}=\{x_{1}\ldots x_{n}\}$, i.e., ${Ø}_{i}^{0}=x_{i}$

**Second**, compute the sum of the products and each output of the hidden layer in the order from *l*_1_ to *l_m−_*_1_ for ‘*k*’, progressing from 1 to m−1

compute $h_{i}^{k}=w_{i}^{\to k}{Ø}^{\to k-1}+b_{i}^{k}$$=b_{i}^{k}+\sum_{j=1}^{r_{k-1}}w_{ji}^{k}{？}_{j}^{k-1}$, for *i =* 1…*r_k_*compute ${Ø}_{i}^{k}=g(h_{i}^{k})$, for *i =* 1…*r_k_*

**Third**, compute the output *y* for the output layer *l_m_*

Compute $h_{1}^{m}=w_{1}^{\to m}{Ø}^{\to m-1}+b_{1}^{m}=b_{1}^{m}+\sum_{j=1}^{r_{m-1}}w_{j1}^{k}{?}_{j}^{k-1}$Compute $Ø={Ø}_{1}^{m}=g_{Ø}(h_{1}^{m})$, where the MLP uses the denotations below.

The $w_{ij}^{k}$ is the weight for perceptron *j* in the layer *l_k_* for the incoming node *i,* $b_{i}^{k}$ is the bias for the perceptron *i* in layer *l_k_*, $h_{i}^{k}$ is a product of some plus bias for perception *i* in layer *l_k_*, ${Ø}_{i}^{k}$ is the output for node *i* in layer *l_k_*, *r_k_* is several nodes in layer lk, $w_{i}^{\to k}$ is the weight vector for perceptron *i* in layer *l_k_,* and ${Ø}^{\to k}$ is the output vector for layer *l_k_.*

#### 2.4.2. The Adaptive Neuro-Fuzzy Inference System (ANFIS)

The ANFIS is developed by combining the ability to learn the neural network and its advantage of a rule-based fuzzy inference system, which enables it to integrate a past observation into the process of classification [27,28]. This combination makes ANFIS a good model for overcoming the limitations of individual modelling. Jang JS, in 1997, described the ‘defuzzifier’, ‘fuzzifier’, and ‘fuzzy’ databases as the three parts of a fuzzy system [28]. Even though they are different from each other, the well-known fuzzy inference systems are Mamdani’s system [29], Tsukamoto’s system [30], and Sugeno’s system [31].

The ANFIS architecture contains five layers: layer 1 is the input layer, layer 2 is the input membership function (MFs), layer 3 is the association rules, layer 4 is the output membership function, and layer 5 is the model output (see Figure 3). After the construction of the fuzzy system, it specifies the relationship between the fuzzy variables using the ‘if-then’ fuzzy rules. The first order of Sugeno’s system has the following rules, considering that the FIS contains a single output (*f*) and two inputs (*x* and *y*):$$Rule\;\left(1\right):\;if\;\mu \left(x\right)\;is\;A_{1}\;and\;\mu \left(y\right)\;is\;B_{1},\;then\;f_{1}=p_{1}x+q_{1}y+r_{1}$$
$$Rule\;\left(2\right):\;if\;\mu \left(x\right)\;is\;A_{2}\;and\;\mu \left(y\right)\;is\;B_{2},\;then\;f_{2}=p_{2}x+q_{2}y+r_{2}$$
where *A* and *B* are membership functions, and *p*, *q*, and *r* are parameters for the outlet functions. Assuming these parameters, the structure of the ANFIS with five layers is as follows:

**Layer 1**: every *i*’s node is an adaptive node that has the following node function on this layer:$$Q_{i}{}^{1}=\mu_{Ai}\left(x\right)\;for\;i=1,\;2\;or\;Q_{i}{}^{1}=\mu_{Bi}\left(x\right)\;for\;i=3,4$$

Where *Q_i_^1^* is for input *x* or *y*, that is, the membership grade. Here, the Gaussian membership function was selected due to the fact that it has the lowest error of prediction.

**Layer 2**: In this layer, the ‘T-Norm’ operator connects every rule using the ‘AND’ operator between the inputs and is presented as:$$Q_{i}{}^{2}=w_{i}=\mu_{Ai}\left(x\right).\;\mu_{Bi}\left(y\right)\;for\;i=1,2$$

**Layer 3**: In this layer, the output is the ‘Normalized firing strength’, and the labelled norm for every neuron is as follows:$$Q_{i}^{3}=\bar{w}=\frac{w_{i}}{w_{1}+w_{2}}\;\;\;1,2$$

**Layer 4**: In this layer, every *i*’s node is an adaptive node and executes the consequence of the rules by considering *p*_1_, *q*_1_, and *r*_1_ as an irregular parameter, as follows: $Q_{i}{}^{4}=\bar{w}\left(p_{i}x+q_{i}y+r_{i}\right)=\bar{w}f_{i}$

**Layer 5:** In this layer, the overall output is calculated by summing all the incoming signals, as follows:(1)$$Q_{i}{}^{5}=\bar{w}\left(p_{i}x+q_{i}y+r_{i}\right)=\sum w_{i}f_{i}=\frac{\sum w_{i}f_{i}}{\sum w_{i}}$$

#### 2.4.3. The Support Vector Machine (SVM)

According to A. M. Kalteh, the support vector machine (SVM) may be utilized for both prediction and classification [11,32]. The type of regression of this model is known as support vector regression. It is used to define regression using SVM and structural risk reduction. Figure 4 depicts the framework of the SVM regression technique that can simulate non-linear situations in the real world. The estimation obtained using this regression is important for estimating a function of the given dataset:(2)$${\left(x_{i}d_{i}\right)}_{i}^{n}$$
where *x_i_* is the input vectors, *d_i_* is the actual values, and n is the total number of the dataset. Hence, SVM has the following regression function:y=f(x)=ωϕ(xi)+b
where *φ* is a non-linear mapping function, and ‘*ω* ‘and ‘*b***’** are parameters of the function of the regression that can be determined by assigning positive values for the slack parameters of *ξ* and *ξ** and the minimization of the objective function, considering ‘*c*’ as the regularized constant and $\frac{1}{2}||w|{|}^{2}$ as the weight vector norm, as shown below. 

The Minimization:$$\frac{1}{2}||w|{|}^{2}+c\left(\sum_{i}^{n}\left(\xi_{i}+\xi_{i}*\right)\right)$$

This is subjected to:$$\left\{ \begin{array}{l} w_{i}\phi (x_{i})+b_{i}-d_{i}\leq \varepsilon +\xi^{*}\\ d_{i}-w_{i}\phi (x_{i})+b_{i}\leq \varepsilon +\xi^{*},i=1,2,\ldots n\\ \xi \xi^{*} \end{array}\right.$$

The optimization problem stated above could be improved to obtain a dual quadratic problem of the optimization, defining the lag-range multipliers *α_i_* and *α_i_**. In addition, the vector ‘*w*’ can be computed by identifying the optimization solution problem, as follows: $$w*=\sum_{i}^{n}(\alpha_{i}-\alpha_{i}*)\varphi (xi)$$

Hence, the SVM regression function is changed to:$$f(x,\alpha_{i},\alpha_{i}{}^{*})=\sum_{i=1}^{n}\left(\alpha_{i}-\alpha_{i}{}^{*}\right)k(x_{i},x_{j})+b$$
where *b* is the bias term and *k (x_i_, x_j_)* is the kernel function that can be expressed as:$$k(x_{1},x_{2})=\exp(-\gamma ||x_{1}-x_{2}|{|}^{2})$$
where γ is the parameter of the kernel.

#### 2.4.4. The Multiple Linear Regression (MLR)

MLR is commonly used as a statistical modelling technique to observe the linear relationships between numerically measured variables. It is a form of linear regression used to examine the linear relationship between a single target variable and several input variables. In this technique, the dependent variable (Y) is supposed to be affected by the independent variables (Xi), and the estimated model can be expressed as:(3)$$y=\beta_{0}+\beta_{1}x_{1}+\beta_{2}x_{2}+\ldots +\beta_{n}x_{n}+\varepsilon \;\;\;\;\;\;i=1\mathrm{to}n$$
where *y* is the target variable,$\beta_{0}$ is the regression constant, $\beta_{i}$ are coefficients of the input variables, and $x_{i}$ is input variables.

### 2.5. Ensemble Modelling

In the AI industry, ensemble modelling is computed by combining the estimated predictions of multiple single AI-driven models. This combination can advance the final model’s performance and it can provide better predictions than the individual models [33]. To boost the performance of single models, this study used two linear ensemble techniques, the weighted average ensemble (WAE) and simple average ensemble (SAE), and two non-linear ensemble techniques, the ANFIS ensemble (ANFISE) and neural network ensemble (NNE), were applied (see Figure 5). These ensemble techniques have been applied in various studies for purposes such as the clustering and classifications of medical data, web ranking, economic forecasting, etc. [34,35,36,37,38,39]. Considering this situation, this study also applied the ensemble technique to predict COVID-19 mortality in East Africa.

#### 2.5.1. The Linear Ensemble Approaches

In this approach, the simple average (SA) and weighted average (WA) ensemble techniques were applied. In the simple average technique, the arithmetic average of the output, the COVID-19 mortality, of every single AI-driven model is taken as the final predicted mortality in the region. Meanwhile, in the weighted average technique, the prediction is computed by assigning weights to each output relative to its importance.

The formula for the simple average: $COVID\text{-}19=\frac{1}{N}\sum_{i=1}^{N}COVID\text{-}19_{i}$, where COVID-19 is the output of the SA ensemble model, and *COVID-19_i_* is the output of *i*th single AI-driven model. The formula for weighted average: $COVID\text{-}19=\sum_{i=1}^{N}w_{i}COVID\text{-}19_{i}$, where *w_i_* is a weight for the output of *i*th method and is computed using the performance measure called the determination coefficient (DC) and can be calculated with $w_{i}=\frac{DC_{i}}{\sum_{i=1}^{n}DC_{i}}$, where *DC_i_* is the coefficient of determination for the *i*th model.

#### 2.5.2. The Non-Linear Ensemble Approaches

In this approach, the non-linear averaging was computed by training the single AI-driven non-linear models (FFNN and ANFIS) using the COVID-19 mortality values predicted by these single models, and the neural network ensemble (NNE) and ANFIS ensemble (ANFISE) were applied. In NNE, the non-linear averaging was performed by training different FFNNs by feeding the output of an AI-driven single model as an input. Then, the maximum epoch number and neurons of the hidden layer were determined by trial and error. Meanwhile, in ANFISE, the predicted COVID-19 mortality based on the AI-driven single model is fed to ANFIS for training using a different number of epochs and membership functions (MFs).

#### 2.5.3. Normalization and Evaluation of Models

Both the target and input data should be standardized before training the model at an early stage to reduce dimensions and guarantee that all the variables receive equal attention [40]. The following normalization formula should be performed on the dataset to established the values within the range of 0–1:$$COVID\text{-}19_{n}=\frac{{\left(COVID\text{-}19\right)}_{i}-{\left(COVID\text{-}19\right)}_{\min}}{{\left(COVID\text{-}19\right)}_{\max}-{\left(COVID\text{-}19\right)}_{\min}},\;i=1\ldots n$$

*COVID*-19*n*, *COVID*-19*i*, *COVID*-19min, and *COVID*-19max represent the normalized, actual, minimum, and maximum COVID-19 mortality values, respectively. Even though the best model for the validation and training steps is determined by trial and error [41], the root mean square error (RMSE) and determination coefficient (DC) are used to measure the performance and efficiency of the developed models. The DC values range from −1 to 1, and a model value approaching 1 yields better results. In addition, the model with lowest the RMSE is considered to be the best model. The formulas are as follows:$$\begin{array}{c} RMSE=\sqrt{\frac{1}{n}\sum_{i=1}^{n}{\left({\left(COVID\text{-}19\right)}_{obsi}-{\left(COVID\text{-}19\right)}_{prei}\right)}^{2}}\mathrm{and}\\ \frac{\sum_{i=1}^{N}{\left({\left(COVID\text{-}19\right)}_{obsi}-\overset{}{{\left(COVID\text{-}19\right)}_{prei}}\right)}^{2}\overset{}{}}{\sum_{i=1}^{N}{\left({\left(COVID\text{-}19\right)}_{obsi}-\overset{-}{{\left(COVID\text{-}19\right)}_{obs}}\right)}^{2}} \end{array}$$
where *COVID*-19*_obsi_*, *COVID*-19*_prei_*, $\overset{-}{{\left(COVID\text{-}19\right)}_{obs}}$, and N are the observed COVID-19 mortality value, predicted COVID-19 mortality value, average of the observed COVID-19 mortality values, and number of observations, respectively.

## 3. Results and Discussions

This study used three AI-driven and one classical mode, namely ANFIS, SVM, FFNN, and MLR, respectively. All the models were trained and tested to model the mortality due to COVID-19 in East Africa. In the results section, the descriptive statistics, sensitivity analysis, single AI-driven black-box modelling, and ensemble modelling of mortality due to COVID-19 are successively presented.

### 3.1. Descriptive Statistics

A line graph was used to present the data on the average daily mortality due to COVID-19 in East Africa (Figure 6). In this graph, the three largest numbers of mortality cases observed per single day were 979, 966, and 737 deaths on 5 July 2021, 7 March 2022, and 1 October 2021, respectively. However, more than 200 cases per day were registered successively from July to August 2021, and we can conclude that this period was the peak time of mortality in the region.

The descriptive statistics in Table 2 present the average, maximum, and minimum values for the target and all the input variables of the training (70% of the dataset and n = 584) and verification datasets (30% of the dataset and n = 146) from 1 April 2020 to 1 April 2022. The average mortality due to COVID 19 was (61.03 ± 69.1) for the training dataset, and it was (46.16 ± 83.59) for the verification dataset. The average number of new daily cases was (2783.5 ± 2423) for the training and (5724.66 ± 6522.36) for the verification data, while the rates of confirmed positive cases per day were 0.041 ± 0.022 and 0.05 ± 0.052 for the training and verification datasets, respectively. The daily vaccine coverage and hospital beds per 1000 people, which are also presented in the table, showed that the average daily vaccine coverage was (26,234.2 ± 47,498.4) and (220,514 ± 332,466) for the testing and verification data, respectively. The hospital beds/1000 people were (28.25 ± 3.50) and (20.4 ± 0.5623) for the training and verification datasets, respectively.

### 3.2. Sensitivity Analysis

The careful selection of the most relevant factors to consider as input variables and the correct adjustment of connecting parameters (such as the hidden neurons, number of iterations, and transfer functions) for any AI-driven modelling are crucial steps required to obtain the optimum prediction level. The number of new cases, rate of positive cases, newly vaccinated individuals, number of cardiovascular diseases (CVD), stringency index, GDP per capita in USD, number of smokers, the prevalence of diabetes mellitus (DM), Hospital beds/1000, and population density were selected from the dataset for the sensitivity analysis. Previously, linear sensitivity analytical approaches have been used to select dominant input variables. However, due to the complex non-linear nature of the COVID-19 data, we had to conduct a sensitivity analysis with a non-linear nature. Hence, the non-linear FFNN was conducted to select the dominant input variables for the modelling of COVID-19 in the area.

The sensitivity analysis conducted in this study is presented in Table 3. Accordingly, the four best dominant input variables with the highest DCs selected were the positive rate (first-ranked), hospital beds/1000 (second-ranked), new cases (third-ranked), and the number of vaccinated individuals (fourth-ranked) based on their chronological order. However, the inputs with the lowest DCs (<0.5) were removed in the modelling process.

### 3.3. Single AI-Driven Black-Box Models

The ANFIS, SVM, FFNN, and MLR were trained and tested for each combination of input variables in the modelling process. Hence, results from each model are presented in Table 4.

The FFNN was the first type of AI-driven model used in this study. The Levenberg–Marquardt technique was used to train this model, which had four inputs and one hidden layer, to estimate COVID-19 mortality in East Africa. Identifying the optimal structure (number of hidden neurons) of the model was a key step in obtaining the best results in the FFNN modelling. The possession of too many neurons may result in overfitting, or too few neurons may result in incorrect information. To determine the appropriate structure of the FFNN model, a trial-and-error technique was used. Additionally, this allowed us to analyze the accuracy of the numerous models trained with the variable’s hidden number. As a result, the best model structure (x-y-z) with the greatest prediction outcomes was discovered to be six hidden neurons with four inputs and one hidden layer, which was noted as (4-6-1).

The second type of AI-driven model used in this study was the ANFIS, which assumes a fuzzy notion to manage the unpredictable circumstances of complicated data of a non-linear nature. In the modelling process, Sugeno’s fuzzy inference system with hybrid algorithms was used to calibrate the parameters of the MFs. The Gaussian, triangular, and trapezoidal MFs were investigated using a trial-and-error approach to produce the best estimation result in predicting mortality due to COVID-19. As a result, the ANFIS model with “Gaussian membership functions” trained over 41 epochs offered better prediction results than the other MFs.

SVM was the third type of AI-driven model used in this study. The kernel of the radial basis function (RBF) was used to generate the SVM model for the combination of all the input variables. RBF was chosen because it has fewer turning parameters and performs better than sigmoidal and polynomial models [42]. Finally, the traditional MLR technique was employed to predict the COVID-19 mortality and to compare the predicted result with those of the other three types of AI-driven models. The linear connection (a-b) between the one target variable and the four input variables was determined using this model and noted as (4-1).

The results of the single black-box models in Table 4 show that the ANFIS was the best-performing AI-driven model in predicting mortality due to COVID-19, with the highest DC (0.9273) and lowest RMSE (0.000125) at the verification stage. The second, third, and fourth best models, based on their performance, were FFNN, SVM and MLR, respectively. The daily COVID-19 data are non-linear and dynamic by nature. Hence, the non-linear AI-driven model, ANFIS, was found to be the best model for predicting the data. However, according to the calculated DCs, the MLR was the worst-performing model in predicting the mortality data. These results showed that the best models for predicting data of a non-linear and dynamic nature are the non-linear AI-driven models, such as ANFIS and FFNN, while the linear regression estimation approach was a poorly performing model in predicting the mortality due to COVID-19 in the study area.

According to the findings of the single black-box models, provided in Table 4, utilizing the best-predicting model in this study (ANFIS) might improve the performance of the prediction using FFNN, SVM, and MLR by 7%, 8%, and 13%, respectively. In addition to these statistics, scatter plots and Taylor diagrams were created to depict the performances of the single AI-driven models.

Figure 7 depicts the correlation between the estimated values obtained from several AI-driven models and the observed value using a scatter plot diagram. In this diagram, the estimated performances of the ANFIS, FFNN, SVM, and MLR models are compared in terms of their predictions of COVID-19 in East Africa. As a result, the ANFIS model indicated fewer spread points in the linking and produce better estimated values than the other models. This might be attributed to ANFIS’s capacity to anticipate non-linear data, such as COVID-19 data, as it has a greater DC than the other AI-driven and MLR models. The finding from the diagram supports those of the other analyses and modellings, which showed that the non-linear predicting approaches performed better than the linear predicting approaches. Moreover, the ANFIS was the highest-performing model in predicting COVID-19 in the study area.

In this study, we understood that the single black-box models and the scatter plot diagram proved that the non-linear predicting approaches performed better than the linear approaches. More specifically, among all the models, the ANFIS model was the best-performing predicting approach for the daily COVID-19 data. This finding is similar to the finding of a study conducted on daily suspended sediment load data using the AI ensemble model [43].

### 3.4. Ensemble Models

The AI-driven ensemble model was developed using the estimated outputs from three single AI-driven models (ANFIS, FFNN, and SVM) and one classical regression model (MLR) as the input variables for the ensemble modelling. This model was developed to boost the efficiency of the single models in terms of the prediction capability. Four ensemble approaches (SAE, WAE, ANFISE, and NNE), as novel ensemble processes, were developed to predict COVID-19 in east Africa, and the results are presented in Table 5. Accordingly, the (*a-b*) structure for the SAE was applied to display the numbers of outputs and inputs used for the prediction. The structure (a, b, c, d) was the structure for the WAE that denoted the weights of the FFNN, ANFIS, SVM, and MLR single models, respectively.

The ANFISE was best-performing among all the ensemble model development combinations. This is due to its resilience in integrating both the fuzzy concept and the artificial neural network capability, which provided the present ANFIS framework. The NNE, WAE, and SAE were the second-, third-, and fourth-best predictors of COVID-19 in the study area. The weighted ensemble technique outperformed the simple average ensemble approach. This is because the WAE assigns weights to parameters depending on their relevance.

The Levenberg–Marquardt method was used to train the NNE model, as it applied to the FFNN, and the tangent sigmoid activation function was utilized for the hidden and output layers. The study conducted by Sahoo et al. [44] indicated that the FFNN approach has the fastest convergence ability; hence, it was used more often in this study than the other ANN training techniques. We used a trial-and-error method to determine the correct number of hidden layers and the optimal epoch number. In ANFISE, Sugeno’s fuzzy inference system, using a hybrid training approach, was used to calibrate the membership function parameters comparable to those of the ANFIS single black-box model. As a result, the ANFISE greatly improved the accuracy of the single models.

The comparison of the prediction performances of the ensemble models and single AI-driven models at the verification and training phases is presented in Table 6. In this table, the NNE boosted the predicting performance of the single models FFNN, ANFIS, SVM, and MLR by 5.6, 2.1, 7.1, and 13.4 per cent, respectively. In addition to this, the ANFISE boosted the performance of FFNN, ANFIS, SVM, and MLR by 13, 6.1, 13.9, and 19.3 per cent, respectively. These numbers show that the capacity for the prediction of COVID-19 was increased in the case of the ensemble models rather than the single models, and these findings were compared to the findings of studies conducted in different fields using AI ensemble models [6,14,35,37]. Hence, these findings showed that ensemble models can be applied to the prediction of COVID-19 in the eastern Africa region more effectively than the single AI-driven models. In addition to this, the findings prove that the non-linear ensemble models are more capable than the linear ensemble models. This might be due to the incapability of linear ensemble approaches to undergo another black-box learning process, unlike the non-linear approaches.

According to Taylor K.E. (2001), we can summarize multiple models’ performances in a single diagram that can help us to easily visualize and understand which model performs better [45]. Therefore, four ensemble approaches were assessed using a two-dimensional Taylor diagram, as presented in Figure 8, which coordinates the correlation coefficients (r) and the standard deviations (sd) for both the observed and predicted values of the ensemble models (ANFISE, NNE, WAE, and SAE). The advantage of using the Taylor diagram is that it combines the predicting performances of different models in a single visual display that quantifies the level of resemblance between the observed and the predicted values. It is observed from Figure 8 that the ANFISE was the best approach in predicting COVID-19 in the eastern Africa region, with (r = 0.9852 and sd = 0.0523), and the SAE was the poorest-performing ensemble approach, with (r = 0.9073 and SD = 0.0821).

In addition to the statistical evidence, the Taylor diagram showed that the ‘r’ vs. ‘sd’ coordinate of the ANFIS ensemble approach was closer to the observed value than the rest of the ensemble approaches, and we can see that the coordinate for the SAE was far from the observed value compared to the other ensemble approaches. This closeness showed that the predicted values obtained from the ANFISE were more closely related to the observed value. Hence, this proves that this ensemble model has the best prediction capability among the other models.

## 4. Conclusions

In this work, the capacity of AI-driven models to predict mortality due to COVID-19 in east Africa was investigated. Before predicting the COVID-19 mortality using single AI-driven models, the data were normalized, and a sensitivity analysis was performed to identify the best dominant input variables. When comparing the results from the single AI-driven models, ANFIS outperformed the other models due to its ability to analyze non-linear, dynamic, and complicated processes using the fuzzy concept and neural network idea. Four ensemble techniques were modelled to improve the performance of the single AI-driven models by aggregating the results from each AI-driven model and using the aggregated result as an input for the ensemble modelling. Because of their capacity to handle unpredictable, non-stationarity, and complicated data, the non-linear ensemble techniques (ANFISE and NNE) outperformed the linear ensemble approaches (SAE and WAE). ANFISE was the best-performing ensemble technique, improving the prediction performance of the single AI-driven models by 13, 6.1, 13.9, and 19.3 percent, respectively.

Overall, the outcome of this study demonstrated the potential capacity of ensemble models to predict mortality due to COVID-19. The result obtained from the ANFIS ensemble model demonstrated that aggregating the outputs of separate AI-driven models leads to a better prediction than employing them individually. The study’s weakness was that it only used black-box models to calculate the COVID-19 mortality. As a result, the use of physically based models in the assembly process should be investigated in future research. Furthermore, due to data limitations, this study used two years of daily COVID-19 mortality data. Therefore it is necessary to test these AI ensemble techniques for further observations in future studies.

## Figures and Tables

**Figure 1 diagnostics-12-02861-f001:**
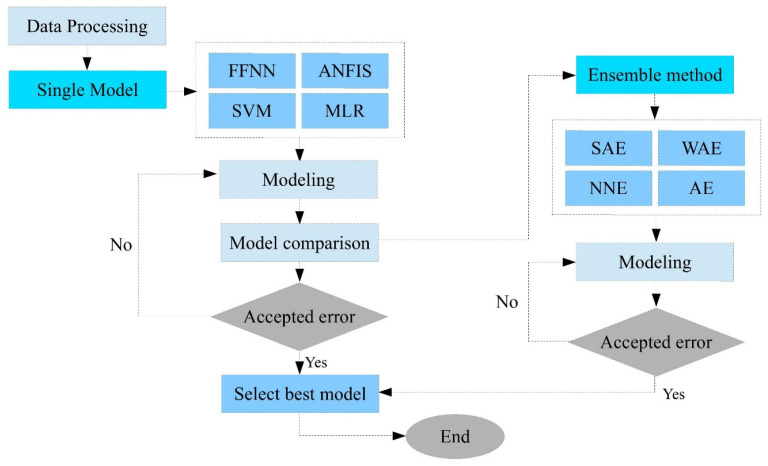
Schematic diagram of the AI-driven models’ development.

**Figure 2 diagnostics-12-02861-f002:**
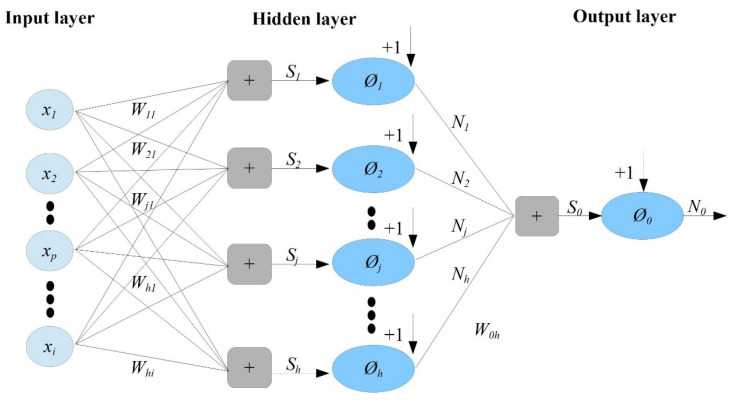
The structure of the FFNN.

**Figure 3 diagnostics-12-02861-f003:**
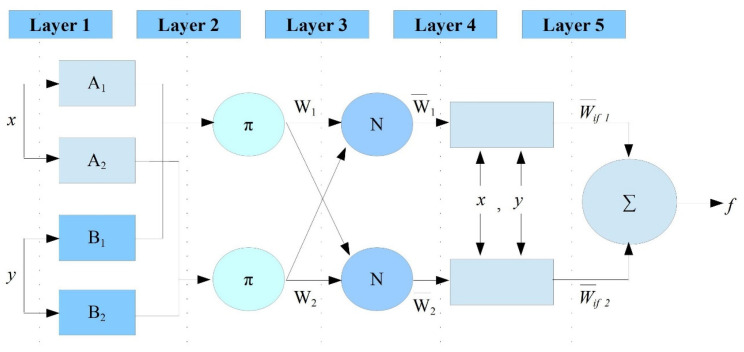
The structure of the ANFIS.

**Figure 4 diagnostics-12-02861-f004:**
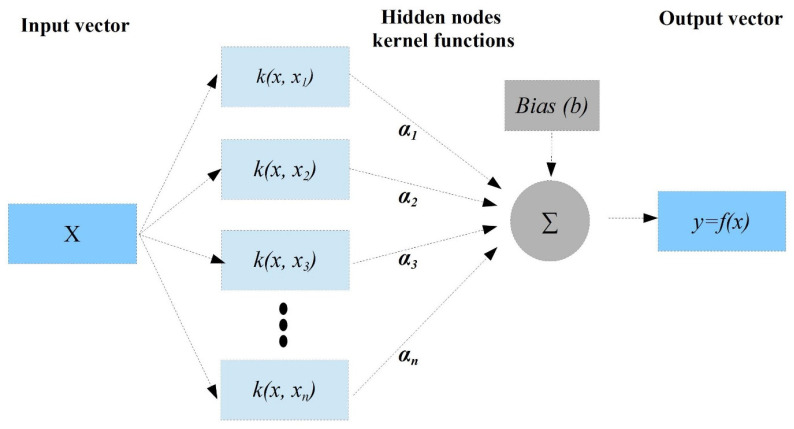
The structure of SVM.

**Figure 5 diagnostics-12-02861-f005:**
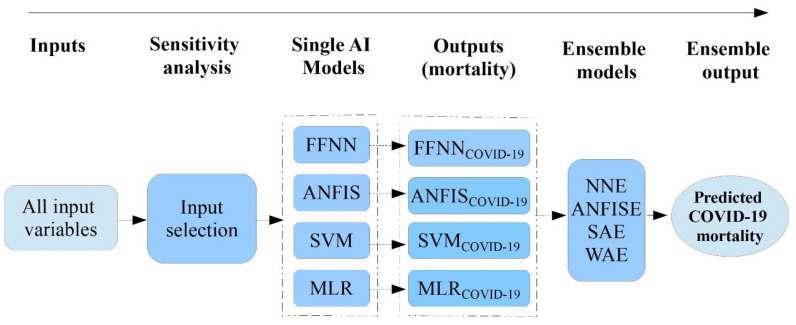
Diagram of the ensemble process.

**Figure 6 diagnostics-12-02861-f006:**
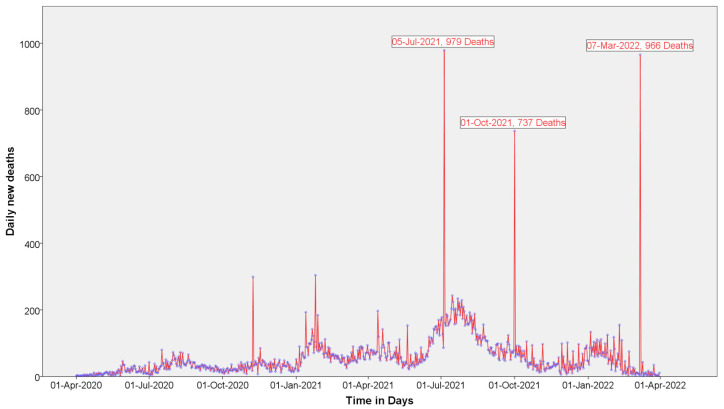
Time series plot for the average daily mortality due to COVID-19 in the region.

**Figure 7 diagnostics-12-02861-f007:**
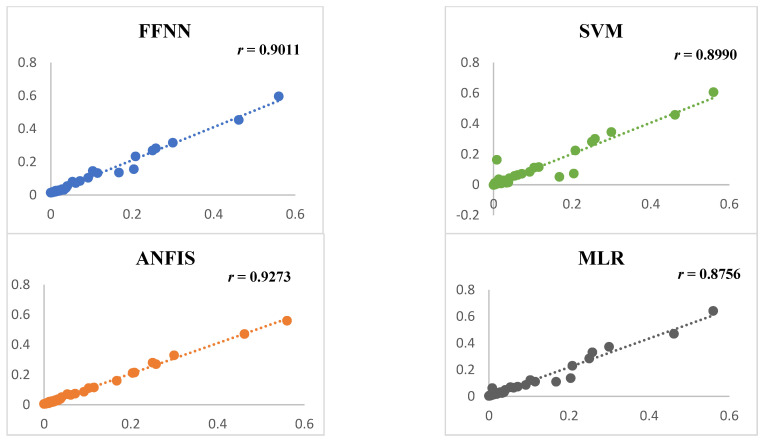
The actual and the predicted COVID-19 mortality rates using four single models at the verification phase.

**Figure 8 diagnostics-12-02861-f008:**
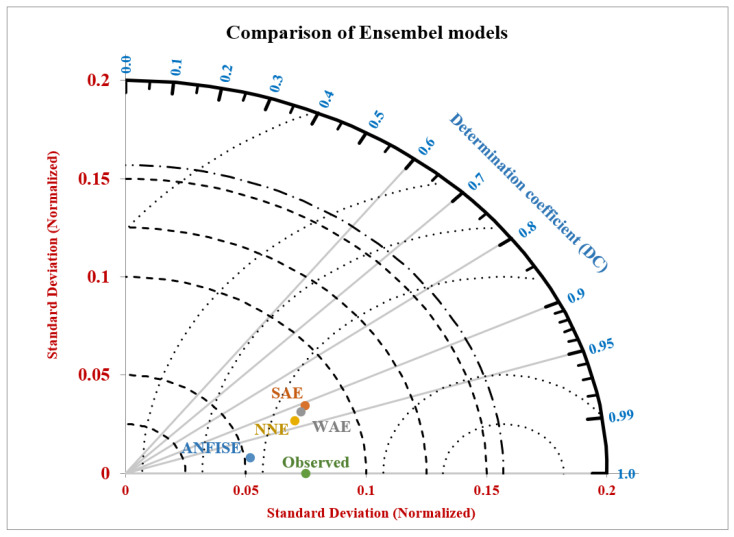
Performance of ensemble models using a normalized Taylor diagram.

**Table 1 diagnostics-12-02861-t001:** The target variable and input variables for this study.

Variables	The Description of Variables
New deaths	New deaths attributed to COVID-19
New cases	New confirmed cases of COVID-19
Positive rate	The share of COVID-19 tests that are positive
People vaccinated	Total number of people who received at least one vaccine dose
Stringency index	A composite metric based on 9 reaction indicators, such as school closures, workplace closures, and travel prohibitions, rescaled to a score between 0 and 100 (100 is the strict response)
GDP per capita/USD	Gross domestic product at purchasing power parity
Number of smokers	Share of male and female smokers
Prevalence of DM	Prevalence of people with diabetes aged 20 to 79
Hospitals beds/1000	Hospital beds per 1000 people
Population density	Number of people divided by land area, measured in square kilometers

**Table 2 diagnostics-12-02861-t002:** Descriptive statistics of COVID-19 mortality and input variables.

Variables	Training Data (n = 584)			Verification Data (n = 146)		
	Min	Mean ± SD	Max	Min	Mean + SD	Max
New deaths	0	61.03 ± 69.1	979	1	46.16 ± 83.59	966
New cases	11	2783.5 ± 2423	27,596	95	5724.66 ± 6522.3	34,125
Rate of positive cases	0.004	0.041 ± 0.022	0.102	0.0	0.05 ± 0.052	0.065
Newly vaccinated	0	26,234.2 ± 47,498.4	276,532	2915	220,514 ± 332,466	1,877,713
Number of CVDs	4655.45	4822.25 ± 17.263	5231.50	4252	4656 ± 0.5268	4986
Stringency index	40.14	51.71 ± 8.80	76.50	29	40.80 ± 2.192	44
GDP per capita/USD	76,254.42	76,321.52 ± 2.35	76,985.23	77,956	76,254 ± 2.589	78,962
Number of smokers	354.2	365.5 ± 56.32	420.5	332.1	354.2 ± 9.536	386.5
Prevalence of DM	6.61	6.71 ± 0.23	6.98	6.51	6.61 ± 0.2652	7.02
Hospitals beds/1000	20.04	28.25 ± 3.50	35.23	18.1	20.4 ± 0.5623	22.6
Population density	2697.26	2725.25 ± 5.62	2756.85	2568.2	2697.25 ± 0.2562	2893.2

**Table 3 diagnostics-12-02861-t003:** Sensitivity analysis used to rank the inputs for the COVID-19 model building.

Inputs	DC	Rank
Positive rate	0.9178	1st
Hospital beds/1000	0.8962	2nd
New cases	0.8617	3rd
People vaccinated	0.8113	4th
Number of smokers	0.2505	5th
GDP per capita/USD	0.2220	6th
Number of CVDs	0.2013	7th
Population density	0.1902	8th
Prevalence of DM	0.0663	9th
Stringency Index	0.0524	10th

**Table 4 diagnostics-12-02861-t004:** Single black-box models for the modelling of COVID-19 using the best input combinations.

Model	Combination of Inputs	Selected Structure	Training		Verification	
			DC	RMSE	DC	RMSE
FFNN	Cases, Pos_rate, vaccine, Hosp_bed	Gaussian	0.8792	0.001478	0.8586	0.001412
ANFIS	Cases, Pos_rate, vaccine, Hosp_bed	4-6-1	0.9146	0.000182	0.9273	0.000125
SVM	Cases, Pos_rate, vaccine, Hosp_bed	RBF	0.8650	0.000210	0.8490	0.000146
MLR	Cases, Pos_rate, vaccine, Hosp_bed	4-1	0.8021	0.000119	0.7956	0.000192

**Table 5 diagnostics-12-02861-t005:** The ensemble approach used to model COVID-19 mortality.

Ensemble Method	Selected Structure	Calibration		Verification	
		DC	RMSE	DC	RMSE
SAE	3-1	0.9446	0.000821	0.9073	0.000245
WAE	0.243, 0.269, 0.249, 0.22	0.9250	0.000123	0.9190	0.000156
ANFIS_E	Gaussian 3	0.9292	0.001658	0.9886	0.000012
NNE	3-6-2	0.9286	0.000120	0.9356	0.000132

**Table 6 diagnostics-12-02861-t006:** The comparison of the prediction level of single AI models vs. non-linear ensemble models.

Ensemble Models	Single Models	Ensemble vs. Single Models	The Difference in Percent (%)	
			Verification	Training
NNE	FFNN	NNE vs. FFNN	5.6%	4.9%
	ANFIS	NNE vs. ANFIS	2.1%	1.4%
	SVM	NNE vs. SVM	7.1%	6.4%
	MLR	NNE vs. MLR	13.4%	12.7%
ANFIS_E	FFNN	ANFIS_E vs. FFNN	13%	5%
	ANFIS	ANFIS_E vs. ANFIS	6.1%	1.4%
	SVM	ANFIS_E vs. SVM	13.9%	6.4%
	MLR	ANFIS_E vs. MLR	19.3%	12.7%

## Data Availability

The data used for the analysis in this study are available from K.H.A. upon request.

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
