# Peer review of "Artificial Intelligence-Driven Ensemble Model for Predicting Mortality Due to COVID-19 in East Africa"

_diagnostics, 2022, doi:10.3390/diagnostics12112861_

Round 1
Reviewer 1 Report
This in an interesting and well-conducted study aimed at evaluating the potential use of AI-driven ensemble model for predicting COVID-19 mortality in East Africa. I suggest expanding the discussion of what has been published on the use of AI-based models as predictors of worse outcomes in this setting. Specifically, there is a huge amount of literature on the use of imaging findings as parameters to be included in these models, given the widespread use of CT and plain radiography to identify, stage, and monitor patients with COVID-10 (i.e. J Pers Med. 2021 Jun 3;11(6):501. doi: 10.3390/jpm11060501.)
1. Introduction is too long-winded.
The paragraph 2.1 seems to be more suitable for the Introduction section rather than the materials and methods. I suggest moving this part to the introduction which, in turn, needs to be summarized not to become too long-winded.
2.2
Please explain why you chose these variables, particularly for what concerns patient’s risk factors considered as negative predictors (smoking, diabetes) since several negative predictors have been established in COVID-19 including cancer, chronic kidney disease, cardiovascular and lung diseases, sarcopenia, and so on (refer to: Radiology. 2021 Aug;300(2):E328-E336. doi: 10.1148/radiol.2021204141).
No major concerns about English language.
Author Response
Reviewer 1:
Comment 1
This is an interesting and well-conducted study aimed at evaluating the potential use of an AI-driven ensemble model for predicting COVID-19 mortality in East Africa. I suggest expanding the discussion of what has been published on the use of AI-based models as predictors of worse outcomes in this setting. Specifically, there is a huge amount of literature on the use of imaging findings as parameters to be included in these models, given the widespread use of CT and plain radiography to identify, stage, and monitor patients with COVID-19 (i.e. J Pers Med. 2021 Jun 3;11(6):501. doi: 10.3390/jpm11060501.)
Response 1: we thank you for your expression about the manuscript and we appreciate the suggestion to expand the discussion part to include similar research like the published manuscript that the examiner recommends. We tried to expand the discussion part immediately after the result is explained with a trach change in the main manuscript, and we included the published manuscripts recommended by the examiner and other manuscripts after a thorough reading about the contents.
Comment 2: The introduction is too long-winded.
Response 2: Dear reviewer, I tried to shorten the introduction part of the manuscript. However, this introduction part includes both the background of the study and the literature review part conducted on COVID-19.
Comment 3: Paragraph 2.1 seems to be more suitable for the Introduction section rather than the materials and methods. I suggest moving this part to the introduction which, in turn, needs to be summarized not to become too long-winded.
Response 3: Dear Reviewer, we accepted this comment and amended the main manuscript with track change as per your comment.
Comment 4: Please explain why you chose these variables, particularly for what concerns patient risk factors considered as negative predictors (smoking, diabetes) since several negative predictors have been established in COVID-19 including cancer, chronic kidney disease, cardiovascular and lung diseases, sarcopenia, and so on (refer to Radiology. 2021 Aug;300(2): E328-E336. doi: 10.1148/radiol.2021204141).
Response 4: Dear Reviewer, the data I have applied to this study was secondary data collected by ‘owid’ and the John Hopkins University (JHU). In this database, there are a lot of variables including the variables you mentioned in comment 3. However, cancer, chronic kidney and lung disease and so on were not collected well and many of them had more than 50% missing data. Hence, I tried to select variables which had some relationship with covid-19 (either + or – relationship) from the literature but I also considered the completeness of variables before including them in the analysis.
Comment 5: No major concerns about the English language.
Response 5: Even though your comment is fine with the English language, we tried to proofread the manuscript once as is pointed out in the track change
Reviewer 2 Report
In this paper authors investigated COVID-19 mortality in east Africa and used AI-driven model. This paper is good in terms of technicality, but presentation is poor. Few of my concerns are as follows:
Please provide the code used to perform experiment. Maybe you can upload on github and share the link (like dataset link).
Many notations are not described before use. May be its good idea to use a table and describe all the notations.
Abstract should be general and should not use abbreviation and too much technical details.
Several typos or grammar mistakes are available e.g. in abstract you wrote Introduction: Please proofread the whole paper.
You can cite few related work such as:
https://www.mdpi.com/1424-8220/22/3/832
https://onlinelibrary.wiley.com/doi/full/10.1111/exsy.13173
Author Response
Reviewer 2:
Comment 1: In this paper authors investigated COVID-19 mortality in east Africa and used AI-driven model. This paper is good in terms of technicality, but the presentation is poor.
Response 1: Dear Reviewer, we thank you for your expression about our manuscript, especially since you explained that our paper is good technically. However, you mentioned that the presentation of this paper is poor. Hence, we conducted a thorough and deep proofread both by the language expert and the Data science experts to improve the poor presentation of the paper. Finally, we authors read the comments from these experts and presented all the changes in track change on the manuscript. Even we added a sub-section about data pre-processing and data analysis, and what software we applied, in the model to make clearer the steps we move in the production of the manuscript.
Comment 2: Please provide the code used to perform experiment. Maybe you can upload it on GitHub and share the link (like the dataset link).
Response 2: Dear reviewer, we would be happy to share codes/commands we applied to the analysis. However, we used a menu-based MATLAB-20 to model single AI models and Ensemble modelling. But we provided a ‘GitHub’ link for the dataset, in the data source section of the manuscript, we used in this study.
Comment 3: Many notations are not described before use. Maybe it's a good idea to use a table and describe all the notations.
Response 3: Dear reviewer, this comment looks more general, however, we tried to explain every detail of formulas and notations immediately after the model is explained in the manuscript.
Comment 4: Abstract should be general and should not use an abbreviation or too much technical details.
Response 4: Dear reviewer, we agreed to the comment you forwarded and we amended by track change to make the abstract a general summary of the manuscript and we minimized the technicality of concepts.
Comment 5: Several typos or grammar mistakes are available e.g. in the abstract you wrote Introduction: Please proofread the whole paper.
Response 5: Dear reviewer, as I mentioned in response 1, the English expert has deeply commented on and corrected the typos and grammatical errors, and even the abstract part is also amended.
Comment 6: You can cite few related works such as:
https://www.mdpi.com/1424-8220/22/3/832
https://onlinelibrary.wiley.com/doi/full/10.1111/exsy.13173
Response 6: we read the two papers recommended by the reviewer for citation, and we found both papers important papers and add some value to our manuscript if included as the citation. As a result, we used the first paper ‘on web development to detect covid-19 by uploading chest x-ray images’ which has a direct linkage with our manuscript.
Round 2
Reviewer 2 Report
Authors updated the paper and no further update requires.